# Development and Characterization of New Green Propolis Extract Formulations as Promising Candidates to Substitute for Green Propolis Hydroalcoholic Extract

**DOI:** 10.3390/molecules28083510

**Published:** 2023-04-16

**Authors:** Andresa Aparecida Berretta, Luana Gonçalves Zamarrenho, Juliana Arcadepani Correa, Jéssica Aparecida De Lima, Giovanna Bonfante Borini, Sérgio Ricardo Ambrósio, Hernane da Silva Barud, Jairo Kenupp Bastos, David De Jong

**Affiliations:** 1Research, Development & Innovation Department, Apis Flora Industrial e Comercial Ltda., Ribeirão Preto 14020-670, SP, Brazil; luana.zamarrenho@apisflora.com.br (L.G.Z.); juliana.correa@apisflora.com.br (J.A.C.); jessica.lima@apisflora.com.br (J.A.D.L.); giovanna.borini@apisflora.com.br (G.B.B.); 2Faculty of Philosophy, Sciences and Letters at Ribeirão Preto, University of São Paulo, Ribeirão Preto 05508-060, SP, Brazil; 3Nucleus of Research in Sciences and Technolog, University of Franca, Franca 14404-600, SP, Brazil; 4Biopolymers and Biomaterials Group, University of Araraquara, Araraquara 14801-320, SP, Brazil; hernane.barud@gmail.com; 5School of Pharmaceutical Sciences of Ribeirão Preto, University of São Paulo, Ribeirão Preto 14040-900, SP, Brazil; 6Ribeirão Preto Medical School, University of São Paulo, Ribeirão Preto 14049-900, SP, Brazil; ddjong@fmrp.usp.br

**Keywords:** Brazilian green propolis, purified extract, soluble dry extract, microencapsulated extract, caffeic acid, *p*-Coumaric acid, artepillin C, baccharin, antioxidant, antimicrobial

## Abstract

The technologies used to produce the different dosage forms of propolis can selectively affect the original propolis compounds and their biological activities. The most common type of propolis extract is hydroethanolic. However, there is considerable demand for ethanol-free propolis presentations, including stable powder forms. Three propolis extract formulations were developed and investigated for chemical composition and antioxidant and antimicrobial activity: polar propolis fraction (PPF), soluble propolis dry extract (PSDE), and microencapsulated propolis extract (MPE). The different technologies used to produce the extracts affected their physical appearance, chemical profile, and biological activity. PPF was found to contain mainly caffeic and *p*-Coumaric acid, while PSDE and MPE showed a chemical fingerprint closer to the original green propolis hydroalcoholic extract used. MPE, a fine powder (40% propolis in gum Arabic), was readily dispersible in water, and had less intense flavor, taste, and color than PSDE. PSDE, a fine powder (80% propolis) in maltodextrin as a carrier, was perfectly water-soluble and could be used in liquid formulations; it is transparent and has a strong bitter taste. PPF, a purified solid with large amounts of caffeic and *p*-Coumaric acids, had the highest antioxidant and antimicrobial activity, and therefore merits further study. PSDE and MPE had antioxidant and antimicrobial properties and could be used in products tailored to specific needs.

## 1. Introduction

In Brazil, at least 14 types of propolis have been identified [1]. Each has unique macroscopic, chemical, and biological characteristics and potencies that vary according to the botanical sources of the materials that bees use to make propolis [2,3,4,5]. In order to provide an option that maintains consistent properties and biological activities, which is essential for research and for health applications, a standardized blend consisting mainly of Brazilian green propolis EPP-AF^®^, derived from material that bees collect from *Baccharis dracunculifolia* (Asteraceae), was developed. Its hydroalcoholic extract, obtained with a specific process and blend of propolis sources, has a reproducible chemical fingerprint and consistent biological properties [4,6,7,8,9,10,11,12]. It contains artepillin C, baccharin, and drupanin, compounds mainly found in *B. dracunculifolia* propolis, along with other common propolis compounds, including caffeic and *p*-Coumaric acids and their derivatives, as well as flavonoids, such as chrysin, aromadendrin-4′-*O*-methyl-ether, and galangin [6,13,14,15,16].

There is substantial evidence for the antioxidant and anti-inflammatory properties of Brazilian green propolis [9,17,18,19,20,21], including inflammasome inhibition [8], kidney protection [22,23], and as an antioxidant [18]. Propolis has also been found to be useful for wound healing, with antimicrobial action [6,10], and as an anti-candida agent [7,11], among others. 

As a consequence of the centuries of traditional use of propolis extracts and the considerable scientific support for various potential biological applications, propolis extracts are already widely used in various regions of the world as a health food or food supplement, and as an over-the-counter product that has anti-inflammatory and antioxidant activities [4,24]. However, although hydroethanolic extracts are still widely used [17,25], safe and efficacious alternative presentations are needed to supply the considerable demand for alcohol-free products [26], including a stable powder form. Dry propolis products are normally obtained through hydroalcoholic extraction and evaporation of the solvent by rotaevaporation, resulting in a soft solid extract, or by lyophilization, resulting in an extremely hygroscopic powder that is difficult to keep dry [8,9,27,28,29].

Propolis is used in various types of products, such as food, food supplements, medicines, and hygienic products, according to each country’s regulations. However, propolis dry extracts prepared with lower cost technologies and that have greater stability and are less hygroscopic would be useful as ingredients in some specific formulations, including capsules, tablets, candies, beverages, and in combination with herbal extracts [24,26].

Therefore, the aim of the present work was the development and characterization of two propolis powder products made from propolis EPP-AF^®^ alcoholic extract (propolis soluble dry extract (PSDE) and microencapsulated propolis extract (MPE)), and a third product comprised of polar propolis fractions (PPF). The physical-chemical profiles and the antioxidant and antimicrobial activities of these propolis products and of their main components were investigated.

## 2. Results

The three extract products differed in their physicochemical aspects and constitutions. Although they were all made from the same blend of EPP-AF^®^ propolis raw material, the extraction process and the technology used to obtain the final formulations of each extract product differed.

MPE was formulated to furnish approximately 40% *w*/*w* of propolis dry matter, while PSDE had 80% *w*/*w*. The quantity of propolis dry matter for MPE was chosen based on information from several previous publications about the microencapsulation process of this product. The highest propolis content using this system had been obtained by Marquiafável et al. [30]. The MPE propolis extract was prepared following the same process, with some slight modifications, which basically involved the exclusion of silicon dioxide, the percentage of propolis, and the encapsulation process. 

In the case of PSDE, although the powder was also obtained using a spray-drying process, the methodology was different from that used for producing MPE. Consequently, it was possible to obtain a powder with approximately 80% propolis dry matter adsorbed in maltodextrin, which resulted in aqueous solubility in a stable physical formulation, though with some impact on the final chemical composition.

The PPF fraction was produced with a different extraction process than the alcoholic extraction used to produce MPE and PSDE. It included maceration in an aqueous alkaline solution at room temperature, followed by acidification to pH 1.0. Subsequently, an extract was obtained using vacuum filtration, which was partitioned with ethyl acetate. PPF is the solid mass obtained after evaporation of this solvent, resulting in approximately 94% *w*/*w* dry material, with no carrier in the composition. In this case, different from the other two extract formulations, the final product yielded only 10% of the mass of the original propolis blend (5 g of raw propolis resulted in 0.5 g of purified extract). On the other hand, alcoholic extraction yielded 42.3% of the mass of the original propolis material. The PPF purification process reduced the yield considerably more, due to precipitation caused by the alkalinization-acidification steps. 

From a macroscopic point of view, both MPE and PSDE powders are very fine, with a light brown color. MPE when dispersed in water at 1% *w*/*v* resulted in an opaque light-brown-amber liquid, while PSDE in water appeared as a transparent brown-amber solution (Figure 1A,B). PPF was not completely solubilized in water at 1% *w*/*v* (a small insoluble residue appeared after the solution preparation) (Figure 1B,C).

Microscopically, MPE was characterized as microcapsules, as previously reported [30], but with improved microcapsule morphology, a more homogeneous aspect, and greater integrity (Figure 2A), probably because of differences in the processing steps, including exclusion of the silicon dioxide ingredient. PSDE was soluble, as expected, due to the hydrolysis procedures used to produce this propolis extract product (Figure 2B), and it did not form microcapsules.

### 2.1. Chemical Characterization of the Propolis Extracts

Chemical characterization of the three propolis products was performed by HPLC analysis and by determination of total phenolic and flavonoid content. They presented different chemical profiles and different total phenolic acid and flavonoid contents.

HPLC analysis results are shown in Table 1 and Figure 3. In the PPF, only caffeic acid and *p*-Coumaric acid appeared in the chromatographic profile. Both biomarkers were at higher concentrations when compared to those found in the other two extracts (*p* < 0.05, Table 1, Figure 3A). Intriguingly, the amounts of caffeic acid and *p*-Coumaric acid were 401.5 and 11.7 times higher, respectively, than those found in the original propolis raw material using the current Soxhlet extraction process required by the normative regulations in Brazil [31]. Artepillin C was not detected in the polar fraction, while in the PSDE and MPE extracts, it was the predominant compound in similar proportions (Table 1). Baccharin was detected only in the microencapsulated extract (Table 1, Figure 3C).

In the PSDE, the following compounds were detected: caffeic acid, *p*-Coumaric acid, 3.5 dicaffeoylquinic acid, 4.5 dicaffeoylquinic acid, drupanin, chrysin, and artepillin C (Table 1, Figure 3B). Its predominant components were artepillin C, drupanin, 3.5 dicaffeoylquinic acid, and caffeic acid. No *p*-Coumaric acid, 3,4-DCQ, or 4,5-DCQ was observed in PSDE (Table 1). Aromadendrin, galangin, and baccharin were also not detected in PSDE.

Caffeic acid, *p*-Coumaric acid, 3,5-dicaffeoylquinic acid, 4,5-dicaffeoylquinic acid, aromadendrin, drupanin, chrysin, galangin, artepillin C, and baccharin were detected in the MPE. The predominant compounds were artepillin C, followed by 4,5 dicaffeoylquinic acid, drupanin, and 3,5 dicaffeoylquinic acid (Table 1, Figure 3C). In the microencapsulated presentation, p-Coumaric and 3,5-DCQ and 4,5-DCQ acids were at higher levels when compared with the other two formulations (*p* < 0.05, Table 1). The composition of MPE was more complex compared to PSDE, presenting all 10 chemical markers that were investigated, thus more closely matching the composition of the original EPP-AF^®^ alcoholic extract. This demonstrates that microencapsulation was superior in maintaining the original propolis complex.

A one-way ANOVA test was used to evaluate the differences between the three extracts, considering the biomarkers caffeic and *p*-Coumaric acids; they were all significantly different from each other (Tukey’s post-test; *p* < 0.05). The quantities of the biomarkers found only in the PSDE and MPE extracts differed significantly (unpaired *t*-test, *p* < 0.05).

An evaluation of the phenolic and total flavonoid amounts, after the normalization of the results (all to 40% propolis dry matter), is presented in Table 2. PPF contained approximately 1/3 of the total phenolics of PSDE and almost 1/2 of the MPE. The total flavonoid content of PPF was equivalent to around 70% of the content found in PSDE and MPE. A larger amount of total phenolics was present in PSDE (*p* < 0.01), while MPE had around 20% less of this group of compounds. PSDE and MPE had similar quantities of total flavonoids, expressed as quercetin. Among the individual compounds that were investigated, all extracts contained caffeic and *p*-Coumaric acids, though in different proportions. 

The results presented in Table 2 were submitted to analysis of variance (ANOVA) followed by the Tukey’s multiple comparisons test (*p* < 0.05). In the analysis of flavonoids, the PPF extract had significantly smaller amounts than the PSDE and MSE extracts. The PSDE and MPE extracts were not significantly different (Tukey’s test). Phenolic acid contents differed significantly among all three extract products (PPF < MPE < PSDE; Tukey’s test, *p* < 0.05).

### 2.2. Antioxidant Activity

According to the procedures adopted for the antioxidant evaluation, lower DPPH values indicate more potent antioxidant activity, while for FRAP values, higher values indicate greater antioxidant activity. PPF was the most powerful extract in terms of antioxidant activity by both methods (one-way ANOVA, Tukey’s post-test, *p* < 0.05). This activity correlated with the caffeic acid content, as this biomarker also had the highest level of antioxidant activity. PSDE and MPE displayed similarly lower DPPH values, based on the FRAP method, though PSDE was more potent than MPE (one-way ANOVA, Tukey’s post-test, *p* < 0.05). Probably this is a consequence of larger amounts of artepillin C in the PSDE extract, which was second to caffeic acid among the propolis components that were evaluated (results validated by both methods). MPE had the least antioxidant activity among the three propolis extracts based on the FRAP method (one-way ANOVA, Tukey’s post-test, *p* < 0.05). Possibly, the microencapsulation process negatively affected the antioxidant determination in this *in vitro* model, as the antioxidant assays may not have had full access to the propolis material in the microcapsules. The *p*-Coumaric acid and baccharin were the least efficient antioxidant compounds (Table 3). Baccharin antioxidant values were not quantified, as the concentrations tested did not give a reaction in these testing procedures.

Caffeic acid, *p*-Coumaric acid, and artepillin C differed significantly (Tukey’s test *p* < 0.05) in antioxidant activity based on both antioxidant evaluation methods.

### 2.3. Antimicrobial Activity

For the microorganism *Staphylococcus aureus*, the PSDE extract showed superior antimicrobial activity (Tukey’s test; *p* < 0.05) when compared with PPF and MPE, which gave similar activities (Table 4). The MPE extract, on the other hand, had the lowest antimicrobial activity against MRSA (*p* < 0.05), while the PPF and PSDE extracts did not differ significantly (ANOVA, followed by Tukey’s multiple comparisons test). All extracts had similar antimicrobial activities against *S. epidermidis* and *K. pneumoniae* (one-way ANOVA, Tukey’s post test) (Table 4).

None of the evaluated standards showed antimicrobial activity in the concentrations evaluated, so it was not possible to compare them. The solvents used to dissolve some of the samples (DMSO and ethanol) did not affect their antimicrobial activity since the minimal bactericidal concentration (MBC) values observed for the solvents were much higher than the MBC found for the extracts.

The PPF had the same MBC against all the strains evaluated, while the PSDE had the most effective antimicrobial activity against *S. aureus* and the worst activity against *S. epidermidis* and *K. pneumoniae*. MPE provided the best antimicrobial activity against *S. aureus* and the least against *K. pneumoniae*.

## 3. Discussion

Natural products differ because of differences in the physical, chemical, and physical-chemical characteristics of their active ingredients, affecting appearance, stability, and performance, as well as their safety and efficacy. In the case of propolis products, hydroethanolic extracts can be inconvenient [17,24], since although this is the best studied and characterized option, there is the inconvenience of containing ethanol, with its strong taste and other intrinsic characteristics. This can be a disadvantage for oral use and when it is used as an ingredient for topical applications, such as in buccal and vaginal creams or gels. Consequently, the development and chemical and biological evaluation of new alternatives for alcohol-free propolis formulations would be useful. Some liquid options have become available that can overcome these problems [17,24,26]; however, concentrated stable options in powder or soft extract forms are still lacking. 

Here, we examined three different alternatives that can be used in various applications, ranging from liquids, creams, and gels to solid forms. From a chemical point of view, the microencapsulated formulation preserved the EPP-AF^®^ hydroalcoholic extract compound profile much better than PPF and PSDE, suggesting that this system would be a useful alternative to the alcoholic liquid form (Table 1, Figure 3). In the chemical analysis, the PPF presented only caffeic and *p*-Coumaric acids, though at high concentrations, while PSDE was found to contain various compounds (caffeic acid, *p*-Coumaric acid, 3,5 dicaffeoylquinic acid, 4,5 dicaffeoylquinic acid, drupanin, chrysin, and artepillin C). Notably, some relevant constituents of the original EPP-AF blend were absent in the PPF and PSDE extracts, including aromadendrin-4′-*O*-methyl-ether, galangin, and baccharin (Table 1, Figure 3). This was a result of the chemical treatments used for their preparation, which resulted in losses during the preparation of the final extract product or their chemical degradation. 

Extraction techniques that use different pHs make it possible to separate molecules by changing their solubility through ionization of organic groups. This strategy was successfully applied in the preparation of the extract products and resulted in products with differing chemical fingerprints, as could be observed in their chromatographic profiles (Figure 3). However, pH changes can also cause chemical modifications, leading to degradation. 

Propolis is a complex mixture of chemical components [3,32]; consequently, understanding in depth all the changes that occur as a consequence of the processing methodologies used for the preparation of the different extracts would be very difficult. However, various relevant details became evident based on the chemical characterization of the extracts, which showed some of the main chemical changes that occurred in their preparation, including loss of dicaffeoylquinic acids and artepillin C, and of the flavonoids galangin, aromadendrin, and chrysin (Table 1 and Figure 3).

Caffeoylquinic acids (CQAs) are unstable and susceptible to thermal and photolytic degradation, and their structures are influenced by pH. They are stable molecules at acidic pH, but they are unstable at neutral and alkaline pH. Under severe extraction conditions, CQAs with a higher degree of esterification progressively degrade to a lower degree of esterification until they form quinic and caffeic acids [33,34]. In the preparation of PPF, the alkalinization–acidification sequence provokes destabilization of CQAs, which explains their non-detection and the increase in the concentration of caffeic acid in this extract, as caffeic acid is one of the degradation products formed by this chemical group of molecules (Table 1A,B and Figure 3). To test this hypothesis, the 3,5 dicaffeoylquinic and 4,5 dicaffeoylquinic acids in the solid residue obtained after the filtration step were quantified. They were not detected in this fraction. In the PPF preparation process, there was considerable reduction in the amounts of these markers due to the conditions used in the extraction; this explains the increase in caffeic acid observed in the PPF.

Little is known about the degradation of artepillin C. Arruda et al. [35] described the formation of isomers through the action of light and temperature, but there is a lack of studies on the effect of pH on the degradation of artepillin C. Artepillin C was absent in the PPF fraction. To further investigate how artepillin C content was affected by processing conditions used for the preparation of the PPF, it was quantified in the solid residue obtained after the filtration step and in the aqueous acid phase obtained after the partition step. Artepillin C was absent in the aqueous acid phase, but it was found in the solid residue. Analysis of the solid residue showed 52.4% recovery of artepillin C compared to the initial content found in the original propolis material. It can be inferred that artepillin C did not totally solubilize or was precipitated after the acidification step, and part of its content was lost as a consequence of the processing methodology used for the preparation of the PPF. We presume that the alkalinization–acidification steps caused some degradation of artepillin C, which should be investigated.

Regarding flavonoid contents, there was a decrease in the amounts of galangin and aromadendrin in the PPF, and differences in the amount of chrysin in the PSDE and PPF. Extraction with pH alterations can change the flavonoid and phenolic contents’ composition of propolis extracts and consequently affect their antioxidant activity. Jurasekova et al. [36] conducted a study of the effect of pH on the degradation of quercetin, fisetin, and luteolin structures, as model molecules representing flavonoids. In alkaline pH, they observed autoxidation of the quercetin molecule, giving rise to molecular fragmentation, resulting in simpler molecular products, and/or dimerization and further polymerization, leading to compounds with a higher molecular weight. Quercetin exhibited high instability in alkaline solution. Considering the mechanism of chemical degradation proposed by these authors and chemical similarities between quercetin and galangin, it can be inferred that the changes in propolis flavonoid content observed in PPF and PSDE occurred due to chemical changes caused by their exposure to alkaline conditions during the extraction process. Studies are needed to confirm these assumptions.

In the PPF and PSDE purification processes, the alkaline pH in the extraction process not only allowed separation of soluble from insoluble components in the alkaline medium, but probably also promoted chemical modifications. Kung et al. [37] reported that the use of alkaline hydrolysis, particularly at pH 8, can improve the quality of propolis by increasing the phenolic flavonoid content. However, at very higher pH values, it caused degradation of flavonoids, but not of the phenolic compounds. In the PPF extract preparation process, acidification of the medium resulted in precipitation of some of the propolis components, which were separated from acid-soluble molecules by filtration and led to a lower yield in terms of mass recovery.

Propolis has potential as a source of natural antioxidants, including phenolic compounds and flavonoids; this characteristic has been hypothesized as one of the reasons for its anti-inflammatory and immunomodulatory effects [22,38]. The antioxidant activity of propolis extracts is a function of their constituents. These compounds differ in structure due to the position and number of hydroxyl groups and substituents on the aromatic ring, which may or may not favor the stabilization of the formed radical. The more substitutions that favor stabilization of the radical that is formed, the greater the antioxidant activity of these molecules [39]. 

The influence of the addition of hydroxyl groups to the aromatic ring can be seen in caffeic and *p*-Coumaric acids, which only differ by one hydroxyl group in the meta-position. As a result, caffeic acid has higher antioxidant values and *p*-Coumaric acid has lower values. Artepillin C showed intermediate antioxidant activity; it has two prenyl groups in a meta position, which can stabilize the radical [39].

The present study showed higher antioxidant activity of PPF due to the high concentration of caffeic acid, followed by PSDE, and weaker antioxidant activity for MPE, due to lower concentrations of artepillin C and caffeic acid. Other studies demonstrated similar behaviors, showing high antioxidant power for caffeoylquinic acid derivatives, artepillin C, and caffeic acid, while other cinnamic acid derivatives (baccharin, *p*-Coumaric acid, and drupanin) did not show antioxidant activity [40,41,42].

There are several methods to determine antioxidant activity [43,44]; some of them are more suitable than others, depending on the nature of the constituents of the sample. The methods are based on two reaction mechanisms: hydrogen atom transfer (HAT), in which the antioxidant agent transfers a proton (H+) to the molecule, and single electron transfer (SET), in which the antioxidant agent transfers an electron to the molecule [43].

Antioxidants are characterized by the presence of electron or hydrogen donating substituents to the molecule or radical according to its reduction potential, the ability to displace the radical formed in its structure, the ability to chelate transition metals involved in the oxidative process, and access to the site of action, depending on hydrophilicity or lipophilicity and the partition coefficient [44]. Phenolic compounds act in different ways as antioxidants; they can donate hydrogen from the hydroxyl group of the phenol to the radical, chelate transition metal, thereby interrupting radical propagation reactions and changing the medium’s redox potential. DPPH is based mostly on the SET mechanism and marginally on HAT. FRAP is totally based on the SET mechanism [43].

The polarity of the solvents influences both mechanisms because it affects the donation of the hydrogen atom and electron transfer. The high capacity of hydrogen bond formation due to polar solvents can drastically alter the antioxidant hydrogen transfer, decreasing antioxidant capacity. pH can also influence this activity since it can, depending on pKa, maintain the molecule in an ionized or a unionized state. Consequently, it is relevant to test the antioxidant activity with more than one method to have more complete information, though the results do not always coincide [43].

The results were submitted to analysis of variance (ANOVA), followed by the Tukey’s multiple comparisons test (α = 0.05). The caffeic acid presented the best antioxidant activity followed by artepillin C, while *p*-Coumaric acid had the least activity. The PPF showed the best antioxidant activity among all extracts by both methods. This was correlated with the caffeic acid content since this biomarker also had the highest antioxidant potential. In Bittencourt et al. [45], the antioxidant effect of propolis extracts was correlated with the phenolic content and consequently with the solvent and extraction method. According to Sawaya et al. [46], the results obtained with the DPPH method were correlated with flavonoid content, while the FRAP method depended on total phenolic and flavonoids content. These aspects may help explain our results, since PSDE contains larger amounts of phenolic compounds compared with MPE, which would explain the antioxidant values produced by the FRAP method, with PSDE > MPE. Based on the FRAP evaluation, MPE had significantly lower antioxidant activity, possibly because it has less artepillin C and fewer total phenolics when compared with PSDE, or because the encapsulation system resulted in a barrier to the delivery of its antioxidant compounds in this in vitro system.

The antimicrobial results found for PPF demonstrated the same MBC values for all bacteria tested (3.44 mg/mL). PSDE was the most efficient against *S. aureus*, followed by *S. aureus* MRSA, *S. epidermidis*, and *K. pneumoniae*. The same sequence of effectiveness of PSDE was found for MPE, though with a greater potency in favor of PSDE. 

Drago et al. [47] evaluated propolis dry extract against some pathogenic microorganisms, and the MBC results obtained against *S. aureus* and *K. pneumoniae* were in the range of 31–125 and >250 mg/mL, respectively. Jansen-Alves et al. [48] compared propolis ethanolic extract, evaporated and solubilized in DMSO with propolis microparticles obtained with pea proteins. The result obtained against *S. aureus* for propolis extract was an MBC of 1.25 mg/mL, while the best results obtained with the microparticles was when the propolis:pea protein proportions were, respectively, 2.5%:2% and 5%:2%, giving values of 20.0 and 5.0 mg/mL. The result obtained here for MPE (3.44 mg/mL) was similar to the result previously published by Marquiafável et al. [30], who reported an MBC of 3.90 mg/mL. Comparing the results for *K. pneumoniae*, the MBC values obtained against the strain tested here were superior to those obtained by Drago et al. [47]. The differences observed between the different samples and publications could be related to the type of propolis and system tested, the microorganism strain [45], the solubility of the extract in the medium [46], and to the compositions of the different types of propolis, since the effect can be a result of synergic activity of the different compounds in combination [45].

The technologies used to produce the three propolis extract products affected their physical appearance, chemical profile, and biological activity. Nevertheless, all three have potential for use in food, food supplements, hygiene and skin care products, and pharmaceutical formulations, according to the specific requirements for each product. 

MPE is a fine powder containing 40% propolis dry matter in gum Arabic, readily dispersible in water (Figure 1 and Figure 2). As a particulate form of propolis, it can easily be used in formulations for solid and semi-solid preparations, such as capsules, pills, and tablets for oral administration, and in creams and gels for topical applications. Presumably, microcapsules would be more stable than the other two formulations, but this requires additional investigation. When propolis mass is microencapsulated, its flavor, taste, and color are less intense, so that it can be used in liquid formulations that require a softer flavor. Microencapsulation preserves the chemical profile of the original propolis extract, a great advantage in maintaining the benefits already known for hydroethanolic propolis extracts. 

PSDE is a fine powder containing approximately 80% propolis dry matter and maltodextrin as a carrier. It is water soluble and could be used in liquid formulations, resulting in a transparent product, though with a bitter taste. Like MPE, it is useful to formulate solid forms, such as capsules, pills, and tablets for oral administration; however, for this product, hygroscopicity of the powder must be taken into consideration because of its effect on shelf life. PSDE showed a distinct composition and antioxidative effect; this extract is a very concentrated powder, including in terms of artepillin C; this can be an advantage, since it can be used in smaller amounts when compared with MPE (half, in fact).

PPF is a purified solid mass of propolis that needs to undergo further processing to become a powder like the other two products. Considering the relatively large concentration of caffeic and *p*-Coumaric acids obtained with this procedure, PPF has potential as a purifying option for these two compounds extracted from propolis. However, further processing steps need to be tested, as these components plus total phenolics and flavonoids account for only 31% of PPF. We were not able to determine the components of the other 69% of PPF with the methodologies that we applied; this could explain the insoluble residue that appeared after dispersion in water. Also, safety studies will be needed to determine if PPF has potential for use in liquid formulations for oral use, as well for use in skincare products, such as gels, creams, and ointments, to take advantage of its considerable antioxidant activity.

## 4. Materials and Methods

### 4.1. Preparation of Propolis Extracts

Propolis raw material was evaluated according to the authenticity and quality requirements published by the Brazilian Ministry of Agriculture (Instruction Normative, no. 3/2001) [31]. Additionally, propolis is routinely analyzed qualitatively and quantitatively by HPLC. A blend of various types of propolis raw material from several regions of Brazil is prepared, though with a predominance of green propolis [6]. For the propolis extract preparation, propolis raw material was initially kept in a freezer at −20 °C for a minimum of 12 h. Next, it was ground and extracted at room temperature with dynamic maceration for 72 h using a hydroalcoholic solution (7:3 ethanol: water), followed by percolation and filtration. This EPP-AF^®^ hydroalcoholic extract was patented by the Brazilian company Apis Flora (Ribeirão Preto, SP, Brazil). In this study, the raw propolis used was in storage for 36 months. The resulting extract was then used as the source for obtaining the microencapsulated propolis extract (MPE) and the propolis soluble dry extract (PSDE). The EPP-AF^®^ blend of raw propolis was also used to produce the propolis polar fraction (PPF). 

MPE was obtained by drying the emulsion prepared with the concentrated hydroethanolic propolis extract and gum Arabic (40:60) using a spray-dryer process, under the conditions published by Marquiafável et al. [30], with modifications in the proportion of propolis, the encapsulant material, and the exclusion of silicon dioxide. PSDE was obtained according to De Andrade et al. [49], with some adjustments, which consisted of the inclusion of an alkaline hydrolysis step for the concentrate obtained by hydroalcoholic extraction, followed by drying of this preparation of propolis together with adsorbent maltodextrin (80:20), with spray-dryer, under the same conditions used for MPE.

To obtain the PPF, the blend of propolis EPP-AF^®^ raw material was stored in a freezer at −20 °C for 12 h minimum and subsequently finely ground and subjected to extraction by maceration in an aqueous solution of NaOH (0.5 M) for one hour at room temperature. Then, this solution was acidified with HCl until the pH reached 1.0. The extract was then vacuum-filtered and partitioned with ethyl acetate. The process was concluded with evaporation of the ethyl acetate.

### 4.2. Chemical Markers

The following isolated compounds were purchased: caffeic acid (*Sigma-Aldrich*, L: SLBZ6416), *p*-Coumaric acid (Sigma-Aldrich, L: 091M119V), and artepillin C (*PhytoLab*, L: 111674647). Baccharin was isolated and identified according to De Sousa et al. [16].

### 4.3. Chemical Characterization of the Propolis Extracts

#### 4.3.1. HPLC Analysis

The three extracts obtained, PPF, PSDE, and MPE, were submitted to high-performance liquid chromatography (HPLC) using Shimadzu equipment with a CBM-20A controller, a LC-20AT quaternary pump, an SPD-M 20A diode matrix detector, and Shimadzu LC software, version 1.21 SP1. The mobile phase consisted of methanol and aqueous formic acid solution (0.1% *v*/*v*), pH 2.7. The method consisted of a 20–95% gradient for 77 min at a flow rate of 0.8 mL/min in a CLC-ODS column (4.6 mm × 250 mm, particle diameter 5 µm, pore diameter 100 A). Detection was set at 275 nm. The chemical markers: caffeic acid, *p*-Coumaric acid, 3,5-dicapheoylquinic acid, 4,5-dicapheoylquinic acid, aromadendrin-4′-*O*-methyl-ether, drupanin, chrysin, galangin, artepillin C, and baccharin were identified and quantified, according to Berretta et al. [6].

#### 4.3.2. Determination of Total Phenol Content

The total phenolic contents of the samples were estimated using a colorimetric assay based on the procedure described by Waterman and Mole [50], with some modifications. PSDE and MPE were weighed and dissolved in 30 mL of water and 30 mL water: methanol (3:2), respectively, in a 50 mL volumetric flask, and then homogenized in an ultrasound bath. Subsequently, the flask volume was completed with the same solvent and filtered through an analytical filter paper. PPF was weighed, dissolved in 5 mL of reagent grade methanol, and homogenized. Then, 1.0 mL aliquots of the samples were transferred to 50 mL volumetric flasks containing 30 mL of water. The reaction with 2.5 mL of Folin-Denis reagent and 5.0 mL of 35% *w*/*v* sodium carbonate was run, and the volume of the 50 mL volumetric flasks was completed with purified water. After 30 min protected from light, the samples were read in a spectrophotometer at 760 nm, using all previous reagents (except samples) as a blank.

#### 4.3.3. Determination of Total Flavonoid Content 

An aluminum chloride colorimetric assay was used to determine total flavonoid content, based on Funari and Ferro [51], with some modifications. PPF and PSDE were weighed and dissolved in 5 mL of reagent grade methanol, and MPE was weighed and dissolved in 5 mL of water:methanol (1:1) in a 10 mL volumetric flask. After the samples were homogenized in an ultrasound bath, the flask volume was completed with the same solvent and filtered through an analytical filter paper. Then, 1.0 mL aliquots of the samples were transferred to 25 mL volumetric flasks containing 15 mL of methanol. The reaction with 0.5 mL of 5% *w*/*v* aluminum chloride was run, and the volume was completed with methanol. After 30 min protected from the light, the samples were read in a spectrophotometer at 425 nm, using a solution of 24.5 mL of methanol and 0.5 mL of 5% *w*/*v* aluminum chloride as a blank.

### 4.4. Antioxidant Evaluation

#### 4.4.1. DPPH Free Radical Scavenging Method

In this method, the free radical is reduced in the presence of an antioxidant molecule, which is observed by a change of color solution, from violet to yellow. The procedure was performed according to methodology described by Lee et al. [52], with some modifications. Mother solutions of PSDE, MPE, and PPF were prepared in ethanol solution 70% *v*/*v* at concentrations of 0.2, 0.4, and 0.1 mg/mL, respectively. Mother solutions of propolis markers were prepared in ethanol 96% *v*/*v*, at concentrations of 15 mg/mL for *p*-Coumaric acid and 0.13 mg/mL for caffeic acid, and in DMSO for artepillin C at a concentration of 2.44 mg/mL. After that, curves of the samples were prepared by adding aliquots of 40, 60, 80 100, and 120 µL for EPP-AF^®^ PSDE, MPE, *p*-Coumaric acid, and caffeic acid; aliquots of 20, 25, 30, 35, and 40 µL for PPF and aliquots of 10, 15, 20, 25, and 30 µL for artepillin C to a microplate and completing the volumes to 200 µL with acetate buffer.

The reaction medium was mixed with 0.4 mL of acetate buffer solution, 0.38 mL of ethanol, 20 µL of which concentration of the curve, and 0.2 mL of 200 µM DPPH solution. The samples were maintained in the dark for 45 min, and the absorbance was read in a spectrophotometer at 517 nm. A blank solution was prepared with 0.4 mL buffer solution and 0.6 mL ethanol, and a negative control prepared with 0.4 mL buffer solution, 0.4 mL of ethanol, and 0.2 mL of 200 µM DPPH solution. The results were obtained in IC50, which is the concentration of the sample necessary to reduce 50% of the DPPH solution. The assay was run in triplicate for each sample. 

#### 4.4.2. Ferric Reducing Antioxidant Power Assay (FRAP)

The FRAP assay is based on a change of color of the complex formed with iron ions and TPTZ; the presence of an antioxidant molecule causes ferric to ferrous ion reduction, which changes the solution color. The procedure follows the Benzie and Strain [53] methodology, with some modifications. 

For the extracts, a mother solution of 0.4mg/mL of ferrous sulfate heptahydrate was prepared and diluted with 3 parts of water, the volume was completed with ethanol, and a standard curve was prepared with water by dilution in the mother solution at concentrations of 10, 20, 30, 40, and 50 µM. The samples were prepared in ethanol solution 70% *v*/*v* at concentrations of 0.2, 0.4, and 0.05 mg/mL for EPP-AF^®^ PSDE and MPE, and with PPF, respectively. Sample concentrations were calculated based on absorbance values obtained from the standard curve. 

For propolis biomarkers, a mother solution of 0.4 mg/mL of ferrous sulfate heptahydrate was prepared in water, and a standard curve was prepared in methanol at the same concentrations indicated above. The samples of caffeic acid and *p*-Coumaric acid were prepared in methanol at concentrations of 0.01 and 0.2 mg/mL, respectively. A solution of 2.44 mg/mL of artepillin C was prepared in DMSO, and an aliquot of 41 µL was transferred to a 1 mL volumetric flask, completed with methanol.

The reaction medium was mixed with 70 µL of the samples and 930 µL of FRAP reagent. The samples were maintained in a 37 °C water bath in the absence of light for 30 min. After that, the samples were read in a spectrophotometer at 593 nm. A blank solution was prepared with 70 µL of water and 930 µL of FRAP solution. The results were expressed in µmolFe^II^/mg sample.

### 4.5. Antimicrobial Activity Determination—Broth Microdilution Method

The samples were evaluated against the gram-positive strains of *Staphylococcus epidermidis* (ATCC 14990), *Staphylococcus aureus* (ATCC 6538), *Methicillin Resistant Staphylococcus aureus* (ATCC 43300), and gram-negative *Klebsiella pneumoniae* (ATCC 10031).

The procedure followed the CLSI and NCCLS methods (CLSI, 2019; NCCLS, 2003). The strains were seeded in Mueller-Hinton agar and incubated at 35 ± 2 °C in aerobiosis for 18 to 24 h. A portion of the colonies was transferred to a sterile sodium chloride solution 0.85% *w*/*v* until the turbidity was equivalent to a 0.5 McFarland standard dispersion. Then, 1 mL of the suspension was transferred to a Falcon tube containing 19 mL of MH broth. 

A mother solution of each biomarker was prepared. Artepillin C was dissolved in DMSO, while p-Coumaric and caffeic acids were dissolved in ethanol, at concentrations of 2 mg/mL for all biomarkers. An aliquot of 0.05 mL of mother solution was transferred to a 1 mL volumetric flask, which was then filled with MH broth. The MPE and PPF were prepared in MH:EtOH (1:1) at concentrations of 275 and 58.5 mg/mL, respectively (corresponding to 110 and 55 mg of propolis/mL). PSDE was prepared in MH at a concentration of 137.5 mg/mL (equivalent to 110 mg of propolis/mL).

Aliquots of 200 uL of the samples were transferred to the first well of the series of a 96-well microplate and 100 uL of MH broth in the other wells; after that, serial dilutions were made by transferring 100 uL of the first well to the second and so on. Then 10 uL of the strain suspension was added to each well, and the plate was incubated at 35 ± 2 °C in aerobiosis for 18 to 24 h. Then, 15 µL from each well was transferred to MH agar petri dish plates, incubated at 35 ± 2 °C in aerobiosis for 24 h, and the MBC was determined. 

### 4.6. Statistical Analysis

After three independent replicate determinations for the measurements for each analysis, tests for One-Way analysis of variance (ANOVA) and Tukey’s multiple comparisons test and unpaired t-test were conducted with a significance level of 0.05. Statistical calculations were conducted with the software GraphPad Prism 6.0.

## 5. Conclusions

We obtained three different propolis extract presentations from the same propolis raw material using different extraction and purification processes and technologies. PPF, a purified soft solid mass of propolis, was found to be rich in caffeic and *p*-Coumaric acids and had the most powerful antioxidant and antimicrobial activities; however, additional studies are necessary to evaluate its safety. PSDE is a water-soluble powder containing approximately 80% propolis dry matter. Its antioxidant and antimicrobial effects were more potent than those of MPE. PSDE was the most efficient against *S. aureus*, followed by *S. aureus* MRSA, *S. epidermidis*, and *K. pneumoniae*. The same sequence of effectiveness of PSDE was obtained for MPE, though with a greater potency in favor of PSDE. Probably this is a consequence of larger amounts of artepillin C, which was second to caffeic acid among the propolis components that were evaluated. Aromadendrin, galangin, and baccharin were absent in the PPF and PSDE extracts, suggesting they had suffered some chemical modification during the extract preparation process. MPE is a stable powder containing 40% propolis dry matter, readily dispersible in water. It maintained the same chemical profile as the original propolis extract; however, MPE had the least antioxidant and antimicrobial activity among the three propolis extract formulations, though in the in vitro conditions, the microencapsulation process may have negatively affected the access to the propolis, which probably impacted the antioxidant and antimicrobial determination. In conclusion, the PSDE and MPE extracts developed and evaluated in this study have distinctive chemical fingerprints and biological activities that could be useful for the development of medicines and food supplements with different specific characteristics and applications, while PPF requires further investigation.

## 6. Patents

Patent requests for PPF, PSDE, and MPE are under preparation.

## Figures and Tables

**Figure 1 molecules-28-03510-f001:**
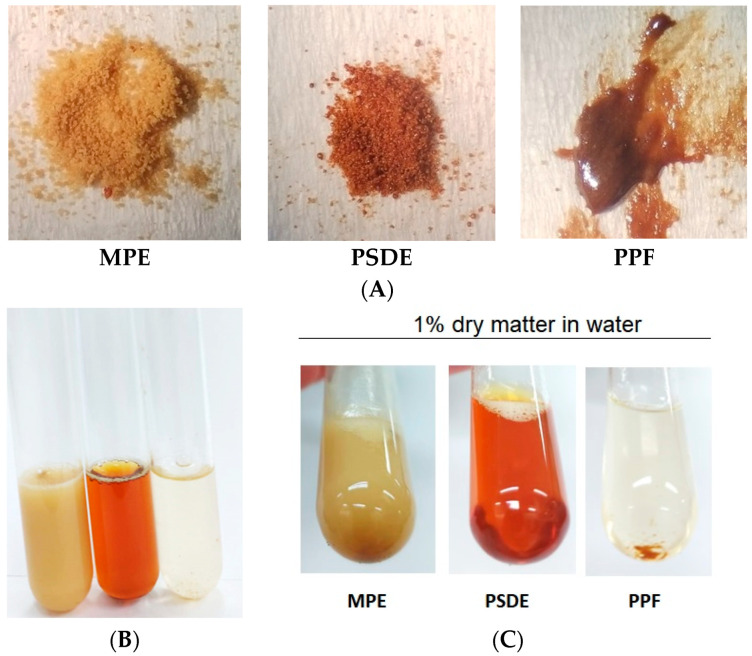
Visual aspects of (**A**) the three extracts from left to right: microencapsulated propolis extract (MPE), propolis soluble dry extract (PSDE), and polar propolis fraction (PPF); (**B**) immediately after mixing in water to obtain 1% *w*/*v* propolis dry matter, in the same order as in a; (**C**) after 30 min, showing precipitates of PPF.

**Figure 2 molecules-28-03510-f002:**
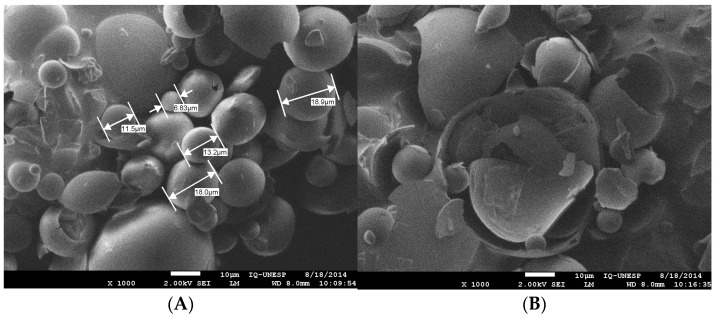
Microscopic aspects of the two propolis extract powders obtained by spray-dryer technology: (**A**) microencapsulated propolis extract (MPE) (40:60 propolis–gum Arabic); (**B**) propolis soluble dry extract (PSDE) (80:20, propolis–maltodextrin).

**Figure 3 molecules-28-03510-f003:**
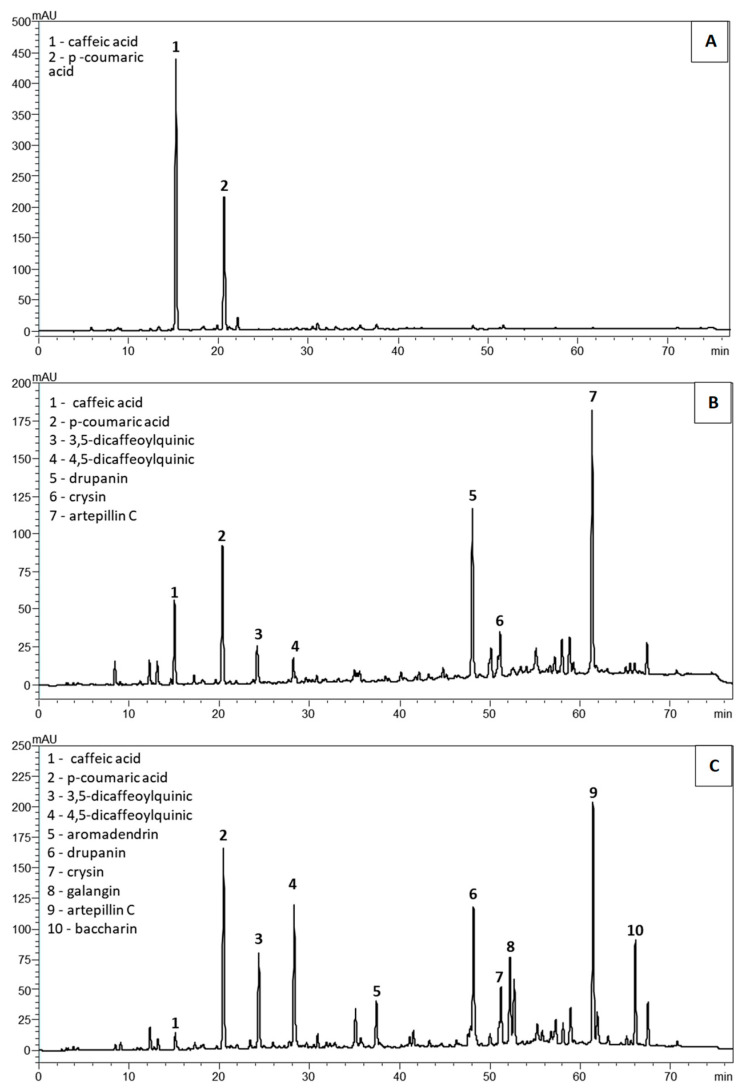
Chromatographic fingerprint of the propolis samples: (**A**) polar propolis fraction (PPF); (**B**) propolis soluble dry extract (PSDE); (**C**) microencapsulated propolis extract (MPE), prepared according to [6]. HPLC analysis was run with a C18 reversed-phase column coupled to a pre-column, with a mobile phase of methanol and an aqueous solution of formic acid (0.1% *v*/*v*), pH 2.7. The method consisted of a 20–95% gradient for 77 min at a flow rate of 0.8 mL/min in a CLC-ODS column (4.6 mm × 250 mm, particle diameter 5 µm, pore diameter 100 A). Detection was set at 275 nm.

**Table 1 molecules-28-03510-t001:** Comparison of the amounts of the components of the propolis extracts (in mg/g) normalized to the same dry matter concentration (40% propolis dry matter).

Constituents	PPF	PSDE	MPE
	mg/g	mg/g	mg/g
Caffeic acid	71.76 ± 7.4	3.52 ± 0.01 *	0.79 ±0.004 **
*p*-Coumaric acid	21.70 ± 1.6	3.49 ± 0.007 *	4.67 ± 0.03 **
3,5 Dicaffeoylquinic acid	-	3.64 ± 0.01	6.62 ± 0.04 ^#^
4,5 Dicaffeoylquinic acid	-	2.37 ± 0.02	11.84 ± 0.2 ^#^
Aromadendrin-4′-*O*-methyl-ether	-	-	2.62 ± 0.04 ^#^
Drupanin	-	9.48 ± 0.07	7.72 ± 0.04 ^#^
Chrysin	-	0.98 ± 0.02	1.22 ± 0.03 ^#^
Galangin	-	-	3.61 ± 0.09 ^#^
Artepillin C	-	22.49 ± 0.5	18.85 ± 0.4 ^#^
Baccharin	-	-	2.58 ± 0.07 ^#^

PPF = polar propolis fraction; PSDE = soluble propolis dry extract; MPE = microencapsulated propolis extract. (*) Significantly different from PPF and MPE (one-way ANOVA, with Tukey’s post-test, *p* < 0.05); (**) Significantly different from PPF and PSDE (one-way ANOVA, with Tukey’s post-test, *p* < 0.05); (^#^) Significantly different from PDSE (Students *t*-test, *p* < 0.05); *n* = 3, mean ± standard deviation.

**Table 2 molecules-28-03510-t002:** Total phenolic and flavonoid compound determination of the three propolis formulations investigated using spectrophotometric methods normalized to the same dry matter concentration (40% propolis dry matter). See Table 1 for definitions of the propolis formulation abbreviations.

Propolis Samples	Total Phenolics (Expressed as Gallic Acid)		Total Flavonoids (Expressed as Quercetin)	
	Content mg/g	RSD (%)	Content mg/g	RSD (%)
PPF	22.58 ± 0.50 *	2.37	16.22 ± 0.60 *	3.53
PSDE	61.62 ± 0.90 **	1.53	22.74 ± 0.30	1.17
MPE	49.45 ± 1.30 ^#^	2.60	23.17 ± 0.60	2.42

RSD: Relative Standard Deviation. (*) Significantly different from PSDE and MPE; (**) Significantly different compared to PPF and MPE; (^#^) Significantly different compared to PPF and PSDE, (one-way ANOVA, Tukey’s post-test, *p* < 0.05); *n* = 3, mean ± standard deviation.

**Table 3 molecules-28-03510-t003:** Antioxidant activity of the propolis extracts and the propolis biomarker components evaluated using DPPH and FRAP methods. (IC50 = concentration of propolis necessary to reduce the oxidation of DPPH by 50%.

Sample	DPPH IC50	FRAP
	(µg/mL)	(µmol Fe^II^/mg)
MPE	10.07 ± 0.37	1.04 ± 0.02 ^#^
PSDE	12.81 ± 0.52	1.51 ± 0.05 **
PPF	4.44 ± 0.16 *	4.33 ± 0.21 *
Caffeic acid	1.11 ± 0.04	32.81 ± 1.72
ρ-Coumaric acid	117.86 ± 5.82	2.53 ± 0.04
Artepillin C	5.29 ± 0.25	4.53 ± 0.09

(*) Significantly different from MPE and PSDE; (**) Significantly different from PPF and MPE; ^#^ Significantly different from PPF and PSDE. (one-way ANOVA, Tukey’s post-test T, *p* < 0.05). Mean ± standard deviation, *n* = 3. Sample (extract) abbreviations defined in Table 1.

**Table 4 molecules-28-03510-t004:** Minimal bactericidal concentration (MBC) determination for the extracts and the compounds selected in this study, followed by the solvents used for the solubilization of some samples according to methodology description.

	MBC (mg Propolis Dry Matter/mL) (Mean ± Standard Deviation)			
Samples	*S. aureus*	*S. aureus* MRSA	*S. epidermidis*	*K. pneumoniae*
MPE	3.44 ± 0.00	6.88 ± 0.02 **	10.31 ± 4.84	20.62 ± 9.67
PSDE	1.72 ± 0.00 *	3.44 ± 0.00	6.89 ± 0.01	6.89 ± 0.01
PPF	3.44 ± 0.00	3.44 ± 0.00	3.44 ± 0.00	3.44 ± 0.00
	µg/mL			
Caffeic acid	>100	>100	>100	>100
*p*-Coumaric acid	>100	>100	>100	>100
Artepillin C	>100	>100	>100	>100
	% *v*/*v*			
DMSO	50 ± 0.00	50 ± 0.00	>50 ± 0.00	50 ± 0.00
Ethanol	25 ± 0.00	25 ± 0.00	25 ± 0.00	18.75 ± 8.84

(*) Different from MPE and PPF, one-way ANOVA, Tukey post-test T, *p* < 0.05; (**) Different from PPF and PSDE, one-way ANOVA, Tukey’s post-test, *p* < 0.05. Mean ± standard deviation, *n* = 3. Abbreviations for the extracts defined in Table 1.

## Data Availability

Not applicable.

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
