# Peer review of "Development and Characterization of New Green Propolis Extract Formulations as Promising Candidates to Substitute for Green Propolis Hydroalcoholic Extract"

_molecules, 2023, doi:10.3390/molecules28083510_

Round 1

Reviewer 1 Report

Respected Authors,

The article is interesting and valuable. Propolis is usually used as a hydroalcoholic extract in medicines or supplements of diet. However, as the Authors underlined alternative formulation of alcohol-free propolis extract is needed. Especially for use in the oral cavity or for children. Therefore, the topic of the article seems important. What’s more, Authors took into consideration the important influence of thermal, photolytic degradation of active compounds of propolis. As the Authors mentioned in the discussion, light, temperature, and pH may influence on destabilization and degradation of active compounds, which causes changes in their biological activity.  

I would like to underline, that the Authors paid attention to this aspect.

We know, that propolis consists of a mixture of substances, so different methodologies used for the preparation of extract may influence their chemical characterization and biological activities such as antimicrobial activity or antioxidant activity.  

The reviewer suggests minor revisions. The list of suggestions and remarks are listed below:

Point 1: I suggest replacing “the extract presentation” with  “formulation of extract” in all text.

Point 2: In the Introduction section, Authors should underline the aim of the work.

Point 3: In the Results section, in line 87 Authors should shortly explain what modifications have been done. Moreover, in this part Authors wrote that: “ PPF was not completely solubilized in water…” so they should discuss this fact in part 3-Discussion.

Point 4: In the Material and Methods section in line 532, the Authors should explain the mentioned modification in the method.

Author Response

Dear Revisor,

Thank you very much for all the contributions presented. We accepted all and revised the manuscript accordingly.

*****

Point 1: I suggest replacing “the extract presentation” with  “formulation of extract” in all text.

Answer: All modified and highlighted in red in the manuscript text.

*****

Point 2: In the Introduction section, Authors should underline the aim of the work.

Answer: The last paragraph of the introduction was changed as presented below:

Therefore, the aim of the present work was the development and characterization of two propolis powder products made from propolis EPP-AF® alcoholic extract (propolis soluble dry extract (PSDE) and microencapsulated propolis extract (MPE)), and a third product comprised of polar propolis fractions (PPF). The physical-chemical profiles and the antioxidant and antimicrobial activities of these propolis products and of their main components were investigated.

******

Point 3: In the Results section, in line 87 Authors should shortly explain what modifications have been done. Moreover, in this part Authors wrote that: “ PPF was not completely solubilized in water…” so they should discuss this fact in part 3-Discussion.

Answer: The clarifications was done, as it is presented below and in the manuscript new version text:

.... The MPE propolis extract was prepared following the same process, with some slight modifications, which involved basically the exclusion of silicon dioxide, the percentage of propolis, and the encapsulation process. 

****

In the last paragraph of the discussion, the revised text:

PPF is a purified solid mass of propolis that needs to undergo further processing to become a powder like the other two products. Considering the relatively large concentration of caffeic and p-coumaric acids obtained with this procedure, PPF has potential as a purifying option for these two compounds extracted from propolis. However, further processing steps need to be tested as these components plus total phenolics and flavonoids account for only 31% of PPF. We were not able to determine the components of the other 69% of PPF with the methodologies that we applied; this could explain the insoluble residue that appeared after dispersion in water. Also, safety studies will be needed to determine if PPF has potential for use in liquid formulations for oral use, as well for use in skincare products, such as gels, creams, and ointments, to take advantage of its considerable antioxidant activity.

*******

Point 4: In the Material and Methods section in line 532, the Authors should explain the mentioned modification in the method.

MPE was obtained by drying the emulsion prepared with the concentrated hydroethanolic propolis extract and gum Arabic (40:60) by spray dryer process, under the conditions published by Marquiafável et al. [30], with modifications in the proportion of propolis, the encapsulant material, and exclusion of silicon dioxide. 

Thank you very much again for helping to improve the quality of this work.

All my best and kind regards,

Andresa A. Berretta

Reviewer 2 Report

I send a review of manuscript ID number Molecules-2335637, of the authors: Andresa Aparecida Berretta, Luana Gonçalves Zamarrenho, Juliana Arcadepani Correa, Jéssica Aparecida Lima, Giovanna Bonfante Borini, Sérgio Ricardo Ambrósio, Hernane da Silva Barud, Jairo Kenupp Bastos and David De Jong New Green propolis extract presentations
as promising ingredients that could substitute green propolis hydroalcoholic extract.

I think that the manuscript is related to an interesting area of scientific research on valuable for the human health preparations prepared on the basis of propolis It should also be noted that an interesting aspect of the work is the preparation of new products, which can have an application use.

The research undertaken by the authors is as relevant and interesting as possible, but the Authors should make a minor revision.

Title of the manuscript:

I suggest changing the title of the manuscript, because it is not very clear, it sounds a bit like a sentence taken from the introduction or conclusions. Please feel free to use the following suggestion, if any:

1/ Study of new/ prepared/ extracts of green propolis....

2/ Development of new extracts... and analysis....

3/ Analysis of new/ prepared green propolis extracts in terms of.../ in the field..

  1. Introduction:

Page 2, line 68 and 69 - propolis presentations? and specific presentations?

Page 2, line 72 - Along this line… - please correct to -  therefore the aim of the work was…

Page 2 lines 72-76 – Please formulate the aim of the work precisely, because in this form is vague and written too generally. Page 2, line 72 - …we developed…, page 10, line 345 - Our study…  - The scientific papers should be written in an impersonal form, please correct throughout all the text.

  1. Results

Page 5, lines 78-98; This part of the work contains partly a description with elements of the methodology. Please correct it.

Page 5, lines 152-154 - In the title of Table 1 - please move the unit [mg/g] to the appropriate place, while the information on the standard deviation and the number of repetitions under the table

Page 5, in Table 1 - Chemical Marker? – Please change this term.

Page 5, lines 173-176; please move the unit [mg/g] to the appropriate place, while the information on the standard deviation and the number of repetitions under the table.

Page 7, lines 212-213- please move the information on the standard deviation and the number of repetitions under the table. Please remove the horizontal line at the beginning of the description of Table 3.

Page 7, lines 228-230 – the same comment as above.

Page 8, line 268 –  pHs?

  1. Conclusions

Pages 14-15; I would like to ask the Authors to select/recommend the best of the three studied preparations in the conclusions summary.

References

All literature is cited in the text.

Author Response

Dear Revisor,

Thank you very much for all comments that helped us to improve the quality of the work.

All points were properly modified in the manuscript text and highlighted in red for better conference. The new version of the manuscript is presented attached.

The points that we not modified were:

"Page 7, lines 212-213- ... Please remove the horizontal line at the beginning of the description of Table 3.

Page 7, lines 228-230 – the same comment as above."

Answer: We are not sure what is meant here. Perhaps this is a difference in formatting in the versions of “word” The Tables appear to be normally organized and formatted in our document..

*****

"Page 8, line 268 –  pHs?"

pHs is plural of pH – see: https://www.usgs.gov/media/images/ph-scale-0

“As this diagram shows, pH ranges from 0 to 14, with 7 being neutral. pHs less than 7 are acidic while pHs greater than 7 are alkaline (basic).”

All the others, can be checked in red in the file (the up-dated version also includes the suggestion of the revisor 1.

Again, we appreciate a lot all suggestions that certainly helped to offer a better quality work in agreement with the quality of Molecules Journal.

All my best, Andresa A. Berretta